# Effects of Partial Replacement of Durum Wheat Re-Milled Semolina with Bean Flour on Physico-Chemical and Technological Features of Doughs and Breads during Storage

**DOI:** 10.3390/plants12051125

**Published:** 2023-03-02

**Authors:** Rosalia Sanfilippo, Michele Canale, Giacomo Dugo, Cinzia Oliveri, Michele Scarangella, Maria Concetta Strano, Margherita Amenta, Antonino Crupi, Alfio Spina

**Affiliations:** 1Research Centre for Cereal and Industrial Crops, Council for Agricultural Research and Economics (CREA), Corso Savoia, 190, 95024 Acireale, Italy; 2Department of Biomedical, Dental, Morphological and Functional Images Sciences (BIOMORF), University of Messina-Viale Annunziata, 98100 Messina, Italy; 3Science4Life S.r.l., Spin-Off of the University of Messina-Via Leonardo Sciascia, 98100 Messina, Italy; 4Agronomic Consultant of AgriCultura Terra di Santo Stefano, C.da Segreto, Santo Stefano di Briga, 98100 Messina, Italy; 5ICQ-RF—Ispettorato Centrale Qualità e Repressione Frodi, Laboratorio di Catania, Via Alessandro Volta 19, 95122 Catania, Italy; 6Research Centre for Olive, Fruit and Citrus Crops, Council for Agricultural Research and Economics (CREA), Corso Savoia, 190, 95024 Acireale, Italy; 7AgriCultura Terra di Santo Stefano, C.da Passo della Scala, Santo Stefano di Briga, 98100 Messina, Italy

**Keywords:** durum wheat, bean, local genotype, physico-chemical characteristics, technological aspects, functional bread, staling process, sustainability, upcycling

## Abstract

The ‘Signuredda’ bean is a local genotype of pulse with particular technological characteristics, cultivated in Sicily, Italy. This paper presents the results of a study to evaluate the effects of partial substitutions of durum wheat semolina with 5%, 7.5%, and 10% of bean flour to prepare durum wheat functional breads. The physico-chemical properties and the technological quality of flours, doughs, and breads were investigated, as well as their storage process up to six days after baking. With the addition of bean flour, the proteins increased, as did the brown index, while the yellow index decreased. The water absorption and dough stability according to the farinograph increased from 1.45 in FBS 7.5%, to 1.65 in FBS 10%, for both 2020 and 2021, and from 5% to 10% supplementation for water absorption. Dough stability increased from 4.30 in FBS 5%-2021 to 4.75 in FBS 10%-2021. According to the mixograph, the mixing time also increased. The absorption of water and oil, as well as the leavening capacity, were also examined, and results highlighted an increase in the amount of water absorbed and a greater fermentation capacity. The greatest oil uptake was shown with bean flour at 10% supplementation (3.40%), while all bean flour mixes showed a water absorption of approximately 1.70%. The fermentation test showed the addition of 10% bean flour significantly increased the fermentative capacity of the dough. The color of the crumb was darker, while the crust became lighter. During the staling process, compared with the control sample, loaves with greater moisture and volume, and better internal porosity were obtained. Moreover, the loaves were extremely soft at T0 (8.0 versus 12.0 N of the control). In conclusion, the results showed an interesting potential of ‘Signuredda’ bean flour as a bread-making ingredient to obtain softer breads, which are better able to resist becoming stale.

## 1. Introduction

Legumes belong to the Fabaceae or Leguminosae family [1], which represents one of the richest in the plant kingdom in terms of species number. Legumes are widely cultivated all over the world [2] and have a strong social-economic impact in both developed and developing countries [1]. They play an important role in human nutrition [3], representing a source of nutrients and bioactive compounds, with high nutritional and health value. Specifically, they are characterized by a high protein content [4], which guarantees a significant supply of essential amino acids such as lysine, leucine, aspartic acid, glutamic acid, and arginine, especially when consumed in combination with cereals. In particular, legume proteins exhibit functional properties, which play an important role in the formulation and processing of foods [5]. They also carry complex carbohydrates, soluble and insoluble fiber, minerals (iron, zinc), and B vitamins [6].

The increased interest in the health aspects and beneficial properties of food has directed research over time towards the formulation of food products enriched with functional components that are able to improve foods from nutritional and health points of view [7]. In particular, the use of different types of flour (rice, soy, buckwheat, and others) has been aimed to replace part of the wheat flour in the formulation of bread [8,9,10]. For some time, there has been a progressive use of legume flour in combination with bread wheat flour or durum wheat semolina for the production of bread and other bakery products, snacks, and pasta, or in terms of purity for the production of gluten-free foods [5]. A study analyzed the use of ‘Pinto’ bean flour for tortilla production [11]. Generally, bread and bakery products lend themselves very well to conveying functional components [7,12], since they are appreciated and consumed at a worldwide level [7], and, almost always, more than once a day. There are several legumes useful for this purpose: chickpeas, beans, peas, and lupins. In particular, bean (*Phaseolus vulgaris* L.) represents 50% of the grain legumes consumed worldwide as a source of human nutrition, and is also widely used for animal feed [13,14]. About 18 million tons are produced in the world, making it the second most import legume in the world, after soy. Over the years, there has been an increase in production; according to FAOSTAT data, about 35 million tons were produced in 2017, an increase of 30% compared to 2007. It is a legume generally consumed as a dry grain, but it can also be found as a canned product or in the form of flour, both in combination with wheat flour and as a 100% replacement of the latter to obtain gluten-free products [14]. As a result, bean flour can be used as a functional ingredient to improve the quality and nutritional value of food products [15]. This study aimed to analyze, from chemical-physical and technological points of view, the effect deriving from the partial replacement of durum wheat semolina with bean flour, through integrations of 5%, 7.5%, and 10%. In particular, a bean genotype called ‘Signuredda’, selected and cultivated in Santo Stefano di Briga, Messina, Sicily (Italy), was used. It is considered a genetic heritage handed down for several generations, and has always been cultivated in this restricted area, but is still not widespread and marketed.

## 2. Results and Discussion

### 2.1. Characteristics of Flours

Wheat re-milled semolina (SC-100%) was supplemented with different percentages of ‘Signuredda’ bean flour at 5% (FBS 5%), 7.5% (FBS 7.5%), and 10% (FBS 10%), according to Różyło et al. and Spina et al. [16,17] for two years (2020 and 2021). 

For the samples, both pure (Figure 1) and the mixes, the protein content and moisture were determined (Table 1). Regarding moisture, it is possible to observe that the integration of bean flour causes a lowering of the recorded values for semolina, as reported in other studies on legumes. It was hypothesized that the influenced was due to environmental and post-harvest conditions, and by the grinding process [18]. In particular, bean flours recorded average values of around 12%, in line with other authors’ findings [19] for red bean flour. Regarding protein content, the control re-milled semolina recorded a value in line with that reported in the literature [12]. ‘Signuredda’ bean flour recorded values significantly higher than the control sample (about 19%), according to Manonmani et al. [20].

The analysis of the nutritional and mineral content of the pure flour of the ‘Signuredda’ bean was carried out during the two years of harvest. Although in the literature there are still no data concerning this genotype, from a nutritional point of view, the recorded results are in line with the study of Marletta et al. [21] on the fresh raw bean seed, although with some minor differences. As shown in Table 2, the data of the ‘Signuredda’ bean for the two years do not show substantial differences when compared with each other. Taking what is reported for raw seeds in the INRAN tables as a reference, we highlight differences mainly concerning the carbohydrate content, which varies from about 43.1 g/100 g of the ‘Signuredda’ bean to 47.5 g/100 g of the raw bean seed. The lipid content varies from 1.4 g/100 g of the ‘Signuredda’ genotype to 2.0 g/100 g of the raw bean, while the protein content ranges from 18.8 g/100 g of the ‘Signuredda’ bean to 23.6 g/100 g of the fresh bean. The fiber content is significant, ranging between 17.1 and 17.3 g/100 g. The data obtained on the bean ‘Signuredda’ are also consistent with those reported in the literature by other authors [22].

Table 3 shows the data relating to the mineral composition of ‘Signuredda’ bean flour for the two years, expressed as mg/100 g, in which a high amount of potassium and phosphorus is observed, while the copper, manganese, zinc, and iron presence is very low. Data on the mineral content obtained on the bean ‘Signuredda’ were higher than those reported by Gomaa and Elhadidy [23].

### 2.2. Color Parameters

The colorimetric parameters were evaluated on the control sample, on the different supplements, and the corresponding bean flour. The results are expressed according to the following parameters: brown index (100-L), red index (a*), and yellow index (b*). The results are shown in Table 4.

Regarding the brown index, significantly different values were observed between the control sample (10.26) and samples obtained with the addition of the bean flour.

In particular, the latter recorded significantly higher values, compared to the control, which means that the integration of semolina with ‘Signuredda’ bean flour resulted in a browner color of the flour compared to semolina.

The bean flour for the year 2021 displayed a higher brown index (14.49) than the flour in 2020 (13.63). Samples obtained with a supplement of 7.5% bean flour recorded the highest values of the brown index. In particular, FBS sample 7.5%-20 recorded the highest value (12.98), while the lowest value was recorded for FBS sample 5%-21 (11.49).

The red index (a*) showed significant differences, both between the control sample and the samples obtained with the addition of bean flour, and between the latter and the 100% bean flour. In particular, the latter, compared with the integrated samples with different percentages, showed much higher values of a*, between 1.08 and 1.29, which are close to those found by other authors for different varieties of beans [24,25].

The data reported in Table 4 also show that the red index values tended to increase as the bean curd concentration increased, and among the samples integrated with bean flour, the FBS10%-20 sample recorded the highest value of a* (−1.23), while the lowest value was recorded by the FBS5%-21 sample (−1.77).

Regarding the yellow index (b*), the highest results were recorded in the sample SC-100% (17.17), in line with what was reported by other authors [26], and differing significantly from 100% bean flour. From the data reported in Table 4, it emerged that the b* values tend to decrease with increasing concentration of bean flour. Among the samples supplemented with bean flour, the FBS sample 7.5%-20 showed the highest b* values (16.75), while the lowest value was recorded by the FBS10%-2021 sample. The yellow index was presumably correlated with the presence of semolina which, in the latter sample, was present in a lower concentration, compared to a higher percentage of bean flour. Therefore, it is possible to hypothesize that by reducing the percentage of semolina, the incidence of the latter on the yellow index decreases (b*).

As can be seen from the analysis of the data presented in Table 4, the colorimetric parameters of the samples to which bean flour was added did not show significant differences between 2020 and 2021.

Table 5 shows a two-factor ANOVA (analysis of variance) of the color characteristics for the years 2020 and 2021.

For almost all variables, except the brown index, the other color parameters were not significant. The brown index remained almost unchanged over the two years.

Table 6 shows the evaluation of colorimetric parameters of the bread samples based on the level of inclusions, as determined by the two-factor ANOVA (analysis of variance). The brown index of the flour increases as the percentage of integration increases, while a* and b* decrease.

### 2.3. Technological Analysis of Flours

In the present study, the technological properties of the samples obtained from a mix of bean flour and semolina at different percentages and the control sample (SC-100%) were evaluated. As shown in Table 7, the samples were subjected to analysis using a farinograph, which describes the water absorption capacity of the dough, and using a mixograph, which makes it possible to evaluate and monitor the development of the dough.

The bean flour spiked samples showed significantly higher water absorption than the SC-100% sample. As reported in Table 7, water absorption increased from 1.45, in FBS 7.5%, to 1.65 in FBS 10%, from 5% to 10% supplementation, for samples of both years, as also found by Bojňanská et al. [27]. The increase in water absorption could be related to the high protein content of the semolina mixtures with bean flour, which results in a higher binding of these proteins with water and, therefore, a high water retention capacity of legume flour [28,29], which consequently influences the amount of water available for the gluten mesh development [30].

This positive trend of increased water absorption following an increase in the percentage of bean flour supplementation has been found by other authors. Hoxha et al. [31] recorded a similar trend, following the integration of bean flour from 10% to 15%, with values ranging respectively from 56.1% to 57.2%. At the same time, however, our data do not coincide with what was found by other authors [18,28], according to which the addition of legume flour to the wheat dough resulted in a significant reduction in the stability of the dough.

It is conceivable that a longer stability time could result in a dough with higher tolerance to mixing and flexibility during mixing [32].

With regard to the dough development time, defined as the time between the first addition of water and the moment at which the dough reaches optimal consistency [33], our study showed a different trend, obtaining slightly lower values for the samples integrated with bean flour compared to SC-100%. This result indicates that dough with a mix of bean flour and semolina potentially takes less time to reach the optimal texture, i.e., 500 U.B., compared to a semolina dough only. At the same time, among the various FBS samples, at increasing levels of added bean flour, the dough development time did not show significant differences, but remained at almost constant values of around 1.5–1.6 min, and the lowest values (1.45 min) in the samples were recorded at 7.5% supplementation. Data reported by our study seem consistent with the results obtained by Anton et al. [11] for some bean varieties. Regarding the analysis with the mixograph, this allowed this team of authors to measure and record the dough’s technological properties and, specifically, its resistance to the kneading process, by providing a curve called a ‘mixogram’. The consistency of the dough is described by the height of the curve and this, together with the width, is related to the proportion of not-aggregated proteins before the mixing peak to the polymeric proteins after reaching the maximum dough consistency [34].

In particular, from Table 7, it is possible to observe a variable trend of the mixing time, which was higher in the samples integrated with bean flour, compared to the 100% semolina sample. Among the FBS samples, there is a tendency to increase as the percentage of bean meal integration increases. In terms of peak height, Table 7 shows an opposite pattern. In fact, the values tend to decrease as the percentage of bean flour integration increases. The lowest percentage of integration, at 5%, is the one that recorded the greatest height values, which were considerably higher even than those of the SC-100% control.

Table 8 shows a two-factor ANOVA (analysis of variance) of the main technological parameters of the two years 2020–2021. All the farinographic variables were not significant. Regarding the mixographic variables, the mixing time showed higher values in 2020, and a greater peak dough height in 2021.

Table 9 shows a two-factor ANOVA (analysis of variance) referring to the different integration percentages of the two years 2020 and 2021. The farinograph parameters, except for water absorption, increase as the integration percentage increases. Regarding the mixograph, as the percentage of integration increases, the mixing time increases, while the peak dough height decreased.

### 2.4. Water/Oil Absorption Capacity

The water absorption capacity is an extremely important parameter since it can significantly affect some characteristics of the bread, such as the product humidity, the retrogradation of the starch, and the subsequent bread ageing [35]. By comparison, the absorption capacity of the oil by the flours is a very important parameter as it can provide useful information regarding the product stability during the storage phase and its possible degradation, due to phenomena related to oxidative rancidity [3]. 

The water and oil absorption capacities were evaluated to verify the processing aptitude of both the SC-100% control sample and the samples to which bean flour were added.

Regarding this property, as shown in Figure 2, the integration of semolina with bean flour initially determined an increasing trend, reaching the maximum absorption value, equal to 1.76 g H_2_O/g d. m., with the FBS 7.5-2020 sample, which had an integration of 7.5%. This is a rather common trend that is observed in semolina samples enriched with an ingredient with a high protein content [30]. Legume proteins contribute significantly to water absorption, consequently influencing the amount of water available for the development of the gluten mesh. Consistent with our results, the 5% supplementation of the FBS 5%-2020 sample recorded the lowest water absorption value, equal to 1.67 g H_2_O/g d. m. It is important to emphasize, however, that this ability could also be influenced by other parameters, as observed by other authors [36,37]. A higher water retention capacity could be due to the presence of a certain amount of fiber in the bean flour, which, due to this water retention capacity, could significantly influence the texture of the food [38].

The 7.5% integration of bean flour achieved a plateau in terms of water absorption, presumably due to the interaction of semolina with the proteins and fiber of bean flour. 

The oil absorption capacity represents an indication of the rate at which the protein binds to fat in food formulations [39]. Many authors [40,41] believe that this parameter could be influenced by particle size, starch, and protein content and type of legume flours. 

The samples obtained from the integration of semolina with bean flour, relating to the year 2020, recorded an increasing trend of absorption as the percentage added increased. The highest value (3.40 g/100 g d. m.) was found in the FBS 10%-2020 sample, supplemented with 10% bean flour. The same integration obtained different data in the years 2020 and 2021, with a reduction in absorption in the FBS 10%-2021 sample, which recorded a value of 2.37 g/100 g d. m. This absorption process is related to the action of the non-polar side chains of proteins, which trap the oil [3,42].

With regard to the water and oil absorption of pure bean flour, it is possible to observe, for both the years 2020 and 2021, a higher predisposition to water absorption. This trend is in line with what was found for other legume flours [3].

### 2.5. Leavening Test

The leavening test was carried out both on the SC-100% sample and on the mix of semolina and bean flour, obtaining the trend shown in Figure 3A,B. The leavening process is characterized by the development of gases, whose production speed is substantially linked to the activity of yeast and the gluten mesh [27]. The most relevant data highlight the presence of significant differences between 2020 and 2021, both in terms of growth and time. The time required to next reach two constant measurements ranged between 40 and 90 min, with considerable differences between the two years. In fact, the 2020 samples recorded a higher increase in leavening capacity than the 2021 samples, taking less time, at most 50 min. In particular, the FBS5%-20 sample had growth of 103.3%, FBS 7.5%-20 had growth of 113.5%, and FBS 10%-20 had growth of 138.2%. As for the 2021 samples, growth stopped at lower values: the FBS5%-21 sample increased by 84.8%, FBS7.5%-21 by 94%, and FBS10%-21 by 98%, in a time interval between 60 and 90 min. A significant result was also obtained by comparing the leavening test of the SC-100% sample with the other samples (trials). The samples obtained from the various supplements of bean flour recorded a higher increase than the control sample, which stopped at a growth of 89.42%. In particular, the increase in the percentage of bean flour, from 5% to 10%, led to a significant increase in the dough leavening capacity. It is therefore evident that the use of bean flour, at different percentages, improved the leavening process and the increase in the volume of the dough, compared to semolina. 

### 2.6. Evaluation of Quality Parameters of Bread

After cooking, the physical properties of both the SC-100% control samples and the mixes were evaluated. The specific volume, specific gravity, porosity, and internal structure of the bread were evaluated, and the results are reported in Table 10. In particular, the specific volume is a parameter that provides information about the baking performance of bread, one of the most important qualitative parameters [5]. Results showed the absence of a significant difference between the SC-100% sample and samples supplemented with bean meal (Table 10). Among all the samples, the FBS 10%-2021 sample was the only one to record a reduction in the specific volume as the percentage of bean flour increased, probably due to the action of starch granules, according to Bhol and Bosco [43]. The other samples showed a different trend, with an increase in specific volume following the increase in the percentage of bean flour, recording a peak with the FBS 7.5%-21 sample, in agreement with other authors [44].

Regarding specific weight, the samples supplemented with different percentages of bean flour recorded slightly lower values than the SC-100% sample, except for FBS 5%-2021, while among the different samples the values were constant. The FBS 5%-2021 sample reported the highest value and FBS 7.5%-2021 the lowest value.

The bread porosity is influenced by mixing and leavening processes [45]. Specifically, the oxygen incorporated in the dough throughout the mixing process causes the oxidation of ascorbic acid into dehydroascorbic acid, actively interfering in the formation of the gluten mesh. The subsequent leavening process tends to produce CO_2_, which becomes trapped between the meshes of the gluten [46], releasing it into the dough. The gluten mesh and the gases trapped in it create (or tend to create) macropores in the dough, leading to the formation of porosity in the bread during cooking [47].

Starting from the SC-100% control sample, with an index equal to 7, which results in a less developed porosity of the crumb, an improvement was observed following integration with bean flour, with breads showing a higher alveolature. The best value of 5 was recorded in FBS 5%-2020 and FBS 7.5%-2020 samples. No irregularities in the internal structure were reported in any of the samples obtained in this study.

The data obtained suggest a positive correlation between the use of Signuredda bean flour and the physical characteristics of the bread, reporting a clear difference from what is reported in the literature by Bresciani and Marti [48].

Table 11 shows the evaluation of the physical properties of the bread samples produced in the two years: 2020–2021. Two-factor ANOVA was not significant.

Table 12 shows the evaluation of the physical properties of the bread samples produced referring to the different integration percentages of two years: 2020 and 2021. Two-factor ANOVA was not significant for specific volume and specific weight.

### 2.7. Bread Color

The color represents an important quality parameter for bread, since it greatly affects the degree of acceptability of the product by consumers. It is important to underline that color does not depend only on factors related to the raw material, such as the presence of pigments, but can also be influenced by other factors [49].

Bakery products, in fact, can be subjected to chemical reactions during cooking, such as the Maillard reaction, which then affects the color of the finished product.

The color was measured in terms of brown index, a*, and b*. The breads showed differences in color that are expressed in the values reported in Table 13. Significant differences were shown in relation to the replacement of semolina with 5%, 7.5%, and 10% of bean flour (Figure 4A,B). Regarding the brown index, it is possible to observe that both the rind and the crumb became progressively darker as the level of bean flour substitution increased. Specifically, the crust had an increasing trend. The FBS 10%-2021 sample, obtained with 10% bean flour, recorded the highest value (66.05), which means that a higher concentration of bean flour results in a darker color of the crust. As also reported by other authors [36,49], this brownish appearance of bread could be related to the significant fiber content of beans, since these legumes represent an adequate source of dietary fiber [50,51]. The browning of breads prepared with legume flour may also be due to the triggering of the Maillard reaction [1]. In particular, Mohammed et al. [52] believe that this phenomenon could be attributed to the presence of amino acids in legumes, and particularly lysine. The presence of lysine could allow the triggering of a Maillard reaction during cooking, in which the reducing carbohydrates react with the side chain of this amino acid, resulting in the formation of reaction products, such as polymerized proteins and brown pigments. For the red index (a*), higher values were recorded in the breads to which bean flour was added compared to the sample SC-100%. The integration of semolina with FBS allowed the production of breads with a higher red color than bread obtained exclusively with semolina, according to Mohammed et al. [53], who recorded a similar trend in breads obtained from a mix of wheat flour and chickpea flour. Contrary to what was recorded for the brown index, as the percentage of bean flour increased, there was a reduction in the red index (a*). The FBS 7.5%-2020 sample, with an integration of 7.5%, showed the highest value of a* (15.62), while the FBS 10%-2021 sample, at 10% bean flour, recorded the lowest value (12.38). A similar pattern was recorded for the yellow index (b*), for which significantly different values were recorded between the samples obtained from the bean flour mix and the control sample. From the data obtained it is possible to observe how the crust acquired a more yellow color than the control sample. In fact, the latter recorded a lower yellow index than all samples mixed with bean flour, except for the FBS10%-21 sample, which recorded an extremely low value, equal to 14.03 ± 0.59. Among FBS samples, the highest value was recorded by FBS 7.5%-2020 samples, while the lowest value was recorded by FBS 10%-2021 samples. The data shown in Table 13 show that the rind of the samples obtained from the mix of semolina with bean flour demonstrated a higher yellow color than the bread obtained exclusively with durum wheat semolina. With regard to crumb color, significant changes were observed between samples added with increasing percentages of bean meal and the control sample. In particular, the use of bean flour resulted in an increase in the brown index (Table 13). In fact, the FBS 10%-2021 bread crumb, obtained with the integration of more than 10%, recorded the highest brown index value (31.70), which translates into a darker coloring of the crumb, if compared with the SC-100% sample (23.78), while the lowest value among the supplements was recorded by the FBS5%-21 sample. Similarly, the red a* index also recorded significantly different values between the control sample and the breads to which bean flour was added. Indeed, compared to the SC-100% sample, which recorded the lowest value, equal to −2.92, the values tended to increase with the increase in the percentage of flour; the highest value (0.75) was recorded for FBS 10%-2021, while the lowest value (−0.49) was recorded for FBS 5%-2021 bread, consistent with other authors [53]. Significant differences were also recorded for b* values between the mixes evaluated in our study and the control sample. The latter recorded a value of b* equal to 19.34, meaning the crumb appears to be more yellow, compared to the samples to which bean flour was added. In the case of mixtures, the values increased with increasing percentage of bean meal, except for sample FBS 7.5%-2021 (11.06), for which b* was lower than that of FBS 5%-2021 (12.30). The highest value (12.27) was recorded for FBS 10%-2021 bread and the lowest value (10.83) for FBS 5%-2020 bread; both always remained below the value of the SC-100% sample. This could be justified by a reduction in the percentage of semolina, which is the component that has a greater influence on the yellow index. As observed for flour, reducing the percentage of semolina in the dough also resulted in a reduction in the yellow index of the finished product. This was also found by Mariscal-Moreno et al. [51], who achieved a significant reduction in b* in samples to which bean flour was added compared to the control sample.

Table 14 shows the evaluation of colorimetric parameters of the bread samples of the two years 2020–2021, as determined by the two-factor ANOVA (analysis of variance). ANOVA was not significant, except for the variable b*. In 2021, the crust showed a higher yellow index.

Table 15 shows the evaluation of colorimetric parameters of the bread samples based on the level of inclusions, as determined by the two-factor ANOVA (analysis of variance). Regarding the crust, only the yellow index was significant. Regarding the crumb, the brown index was not significant. The red and yellow indexes increased as the percentage of integration increased. 

### 2.8. Staling Process

Samples were stored and analyzed to evaluate the evolution of the physical properties during storage. The measurements were carried out immediately after cooking (T0), on the first day after baking (T1), and on the fourth day (T4) and the sixth day of storage (T6). In particular, the moisture content, volume, weight, height, and hardness of the bread were determined (Table 16).

A progressive moisture reduction was observed in all the analyzed samples during the storage period. Table 16 shows significant moisture content at T1 after baking as evidenced by the FBS5%-2020 sample with a significant reduction from T0 to T1, ranging from 38 to 28%, respectively. This decreasing trend was observed in all the analyzed samples and throughout the storage period, as also observed by other authors [18] during the storage of the bread obtained with a mix of legume flour. In particular, in our study, at T6, the results in the samples supplemented with bean flour showed a greater reduction in moisture content than in SC-100%, with the FBS10%-2021 sample recording a reduction of almost 50% compared to the initial moisture value, although remaining at control levels. These results may be attributed to the higher protein content; however, at T0 this supports the fibers, mainly supplied by the bean coat, which would then fail to perform any water retention function over time.

It is possible to observe a statistically significant difference in the moisture values ranging from T0 to T6 during the storage period.

Regarding the volume, it is important to underline that at T0 almost all bread samples obtained from the mix of bean flour and semolina recorded higher values than the SC-100% sample. This means that the use of bean flour contributed to the formation of a more voluminous bread than that obtained using only semolina. In particular, in our study, the most significant increase in volume was recorded using an integration rate ranging from 5% to 7.5%. 

During the storage period, from T1 to T6, as well as for moisture content, a decreasing trend was also observed in bread volume measurement, which was already significant at T1, after one day of storage. The FBS5%-2020 sample recorded the greatest volume reduction at T6, while a smaller decrease was observed with SC-100% compared to all the other samples. 

During the 6 days of storage, a significant difference in bread volume measurement from T0 to T6 was highlighted for all samples, in particular for the FBS5%-2020 sample, with the exception of FBS10%-2020, which did not show significant results. 

The weight and height of the loaf were analyzed as physical parameters. Table 16 shows at T0 that the recorded weight was slightly higher in the FBS samples than in the SC-100% sample, while no significant differences were found between the FBS samples. As observed for other parameters, the storage period resulted in a significative reduction in both weight and height, and a uniform reduction in all samples. 

The evaluation of the staling process carried out on each sample during storage showed statistically significant differences from T0 to T6 for weight, while regarding height, only the FBS10%-2021 sample recorded significant differences.

Regarding the texture, from Table 16 it is possible to observe how at T0 the texture was lower in breads with bean flour than in bread made with semolina alone, indicating a greater softness of these breads compared to SC-100%. In particular, among the FBS bread, the FBS5%-2020 and FBS5%-2021 samples recorded the highest values. During the storage period, a significant increase in the texture of all the samples up to T6 was found which, consequently, showed greater hardness than the breads at T0, in line with what was obtained by Zhang et al. [18]. 

As observed for the other parameters, the statistical evaluation of the texture results also highlighted significant differences for all samples during the storage period.

## 3. Materials and Methods

### 3.1. Materials

Sicilian durum wheat semolina produced by the agricultural cooperative society ‘Valle del Dittaino’ a.r.l. in Assoro (Enna, Italy) (Latitude 37°57.17 N; Longitude 14°44.87 E) was used for the tests, and also used in the production of bread ‘Pagnotta del Dittaino DOP’. This semolina, called ‘re-milled semolina’, was subjected to a greater grinding than that used for the production of pasta, which makes it smaller in grain size, due to a higher hydration rate [54]. Re-milled semolina was integrated with different percentages (5%, 7.5%, 10%) of ‘Signuredda’ bean flour, a pulse grown during the 2020 and 2021 crop years on the farm ‘AgriCultura Terra di Santo Stefano’, located in Santo Stefano di Briga, Messina (Italy) (Latitude 38°10.19 N, Longitude 15°47.96 E).

The soil was prepared with winter work (20–25 cm deep) and 2 subsequent low works. Sowing was undertaken by hand on 25 March 2020 and 20 March 2021, spreading the seed at a depth of about 5 cm in the soil, in rows 100 cm apart. No fertilization was undertaken during sowing. When the second true leaf appeared, fertilization was carried out with a liquid organic nitrogen fertilizer called ‘Fantastic soil’ from organic farming (Cuore di Terra, Cerignola, Foggia, Italy). An irrigation and fertigation system with drippers and 8-day irrigation shifts was installed in April. Only in the case of excessive rains, favorable to fungal diseases, were treatments with wettable sulfur carried out.

The ‘Signuredda’ bean is a local variety of *Phaseolus vulgaris* L. with a determined growth and a creamy-white pod. The seeds are reniform with a firm, nut-colored, white ileum around which there is a reddish ring. The cotyledons are light yellow. It is a bean that cooks easily, and it does not need to be soaked. It is very mellow and tasty, and has various uses: soups, in salads after boiling, ground to obtain flour, etc. Although it does not have a brown integument, it leaves the cooking water dark in color. It is a genotype with a strong bond with the territory of origin (north-eastern Sicily, in the province of Messina), where it is grown in a still-restricted area.

The authors decided to use this local variety of bean because it is easily grindable, thanks to its slightly waxy, light-yellow cotyledons, which give rise to a flour very similar in color to that of semolina, and its firm tegument, which shatters easily. Furthermore, based on the results of preliminary rheological and technological tests on the doughs of the mixes with durum wheat semolina, it was seen that it had good potential for making bread.

### 3.2. Preparation of Flour from Bean Seeds

‘Signuredda’ bean seeds were milled with a Cyclotec type 120 mill (Falling Number, Huddinge, Sweden) equipped with a 500 µm sieve [55].

### 3.3. Determination of Moisture Content

The moisture content of bean seeds, flours and breads was determined by oven drying (Memmert, Milan, Italy) at 103 °C until constant weight, according to the AOAC method 935.25 [56]. The analyses were carried out in triplicate.

The results are expressed as percentage relative humidity, as described by He et al. [57].

### 3.4. Water Binding Capacity and Oil Binding Capacity

The water binding capacity and the oil binding capacity were determined following Kahraman et al. [58] and Du et al. [41], respectively. An amount of 2 g of flour was added to 24 mL of distilled water or sunflower oil and kept under shaking at 20 °C for 60 min. Following a centrifugation at 4200 rpm × 30 min (Heraeus Multifuge X3 FR, Thermo Scientific, Waltham, MA, USA), the solid residue was weighed. The analyses were carried out in triplicate. 

### 3.5. Color Determination

A CR 200 colorimeter (Minolta, Osaka, Japan) was used for color evaluation. The CIELab colorimetric model was adopted by expressing the results according to the coordinates L*, a* (indicating the change from green to red), and b* (indicating the change from blue to yellow) [59,60]. The brown index (100 − L*) was then calculated, which indicates the tendency to darken ranging from 0 to 100. The analyses were carried out in triplicate. 

### 3.6. Farinograph and Mixograph Analyses

The evaluation of the quality of dough in terms of mechanical resistance was carried out using farinographic analysis, which consists of three specific moments:-Phase of absorption of added water and formation of the gluten mesh;-Stability, in which disulfide bonds are broken and reformed continuously;-Gluten mesh rupture and curve slope [61].

The curves were obtained using 300 g of semolina in the dough chamber, at different percentages of integration of the various flour mixes, using the Farinograph of Brabender (Duisburg, Germany), equipped by Farinograph^®^ software, according to the method AACC 54–21.02 [62].

The analyses were carried out in triplicate.

The strength of the dough, also linked to the quantity and quality of gluten, was evaluated using a mixograph which, together with the farinographic analysis, can provide a sufficiently descriptive measurement of the characteristics of the dough under study [63,64,65,66]. The curves were obtained using the National MFG. Co. (Lincoln, NE, USA), following the AACC 54-40.02 method [62]. The analyses were carried out in triplicate.

### 3.7. Leavening Test

The method described by Canale et al. [67] was followed for the leavening test, using the re-milled semolina used for Dittaino PDO (Protected Designation of Origin) bread integrated with the various percentages of ‘Signuredda’ bean flour. 

The following quantities were used for the preparation:(a)Control dough: 50 g semolina; 1.5 g yeast; distilled water;(b)5% dough: 47.5 g semolina; 2.5 g bean flour; 1.5 g yeast; distilled water;(c)7.5% dough: 46.25 g semolina; 3.75 g bean flour; 1.5 g yeast; distilled water;(d)10% dough: 45 g semolina; 5 g bean flour; 1.5 g yeast; distilled water.

The analyses were carried out in triplicate.

### 3.8. Physical Characteristics of Breads with Different Integration Levels of Bean Flour 

Breads were obtained according to the official AACC 10-10.03 methodology, described by Canale et al. [67]. The formulation of each type of bread is reported in Table 17. Re-milled semolina (Cooperativa Agricola “Valle del Dittaino”, Enna, Italy), pure or partly replaced by bean flour, yeast (Lievital, Lesaffre Italia, Parma, Italy), shortening (Gioia, Unigrà srl, Ravenna, Italy), sodium chloride (Italkali, Palermo, Italy), sugar (Decò, Ravenna, Italy), ascorbic acid (Farmitalia, Catania, Italy), and tap water were mixed in an experimental kneader (National Manufacturing Co., Lincoln, NE, USA) at 25°C for 4 min. The dough was leavened in a thermostatic chamber (Giorik, Sedico, Italy), equipped with a steam humidifier (SD/SD series, Carel, Brugine, Italy), at 30–32 °C, 80–82% RH for 2.35 h, poured into metal pans (7 cm width, 12.5 cm length), and leavened again for 50 min at 30–32 °C. This was followed by baking in an electric oven (Giorik, Sedico, Italy) for 18 min at 218 ± 5 °C.

The following physical characteristics were evaluated on the breads obtained: volume, height, weight, moisture, texture, porosity of the crumb, color of the crumb, and crust. The bread volume was determined by the rape seed displacement method according to the AACC 10–05 method [62]. The height of the loaves was measured using a digital caliper (Digi-MaxTM, SciencewareR, NJ, USA). The analyses were carried out in triplicate. The Ohaus Adventure Pro AV2102C digital scale was used for measuring weight.

The Zwick Z 0.5 Röell texturimeter, Ulm, Germany, was used to measure the bread hardness. This was equipped with an 8 mm diameter cylindrical aluminum probe and configured with 50 g pre-load, 100 mm/min pre-load speed, 20%forced shutdown threshold. The resulting peak force was measured in Newtons (N). The central bread slices of each loaf were visually compared with the eight Dallmann reference images representing the cross-section of breads with different crumb structures. Crumb porosity was evaluated based on the 8-degree Mohs scale modified by Dallmann [68] where, for mold breads, 1 indicates non-uniform structure, large and irregular cells, and 8 indicates uniform compact structure, small and regular cells [69]. The analyses were carried out in triplicate. The humidity of breads was determined in the stove at 103 °C up to constant weight and the color of both crust and crumb was measured. The analyses were carried out in triplicate.

### 3.9. Determination of Bread Staling Rate

Bread was stored for 7 days at 25 °C and packed in cardboard. On the day of baking, 1 day, 4 days, and 6 days later, the loaf moisture, volume, weight, height, and hardness were determined to evaluate the stale rate, according to the AACC method 10–10.03 [62]. The moisture of the bread was determined as reported in Section 3.3 and Section 3.8. The volume, weight, and height of the loaf were determined as reported in Section 3.8. Loaf hardness was evaluated using a texture analyzer (Z 0.5 Zwick Röell, Ulm, Germany) equipped with an 8 mm diameter aluminum cylindrical probe and configured with a preload of 50 g, at a preload speed of 100 mm/min, crust breaking point (forced closing threshold (shut-down threshold) = 20%). The resulting peak force was measured in Newtons (N). The analyses were carried out in triplicate.

### 3.10. Statistical Analyses

The statistical analysis was performed using the Statgraphics^®^ Centurion XVI software package (Statpoint Technologies, INC., The Plains, VA, USA). One-factor and two-factor analyses of variance (ANOVA), followed by Tukey’s HSD test (*p* ≤ 0.05, *p* ≤ 0.01, *p* ≤ 0.001), was carried out on all physicochemical, technological, and breadmaking attributes and the bread staling process. The following two factors were considered: 1. the year of production of the bean seeds; 2. the amount of bean flour. A one-factor analysis determined the interaction of the factors studied, while a two-factor analysis analyzed each factor’s influence (or lack of influence) individually. 

## 4. Conclusions

The results of the present study showed that the addition of ‘Signuredda’ bean flour to durum wheat semolina can change the technological characteristics of the doughs, as well as the qualitative characteristics of the bread produced. In particular, the mixes obtained with various integrations of bean flour recorded higher values on the farinograph than those recorded by semolina alone, regarding the water absorption, attributing to the capacity of the legume proteins to absorb a greater amount of water. The development time of the dough was shorter than that required for the semolina, while the stability increased. Regarding the leavening tests, these data demonstrated a greater leavening capacity of the doughs obtained from semolina integrated with bean flour compared to those with semolina alone. Therefore, it is possible to confirm that the use of bean flour favors the leavening process and the increase in the volume of the dough.

The data collected on the finished product, after cooking, indicated the possibility of using ‘Signuredda’ bean flour, partially replacing the durum wheat semolina, for the production of biofortified bread. In particular, it was possible to obtain with the bean flour a bread with good chemical-physical characteristics. This was particularly the case regarding the volume and porosity parameters, which tend to increase as the percentage of substitution increases, making them superior to those achieved with durum wheat semolina bread alone. The crust color was darker in the bread obtained with the highest percentage of integration (10%), while the other percentages were similar to those from bread made with semolina alone.

Our promising results allow us to state that the replacement percentage of 7.5% was the best result, as it recorded the best values for almost all parameters.

Concerning the staling process, even if the breads integrated with bean flour showed a greater decrease in volume and moisture content, our results indicate staling was at the levels of the control bread. The texture values were lower than those of SC-100%, indicating that the breads with added bean flour are softer and more resistant to staling, and therefore keep better and for longer. 

## Figures and Tables

**Figure 1 plants-12-01125-f001:**
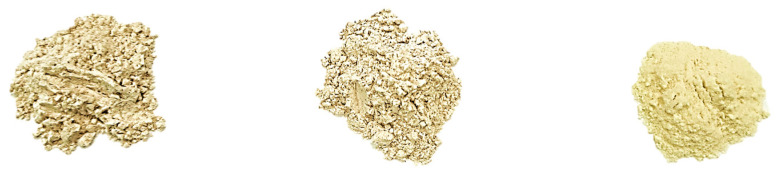
Flours prepared from ‘Signuredda’ bean. From left to right, flours with FBS 100% 2020; FBS 100% 2021; SC-100% (control re-milled semolina).

**Figure 2 plants-12-01125-f002:**
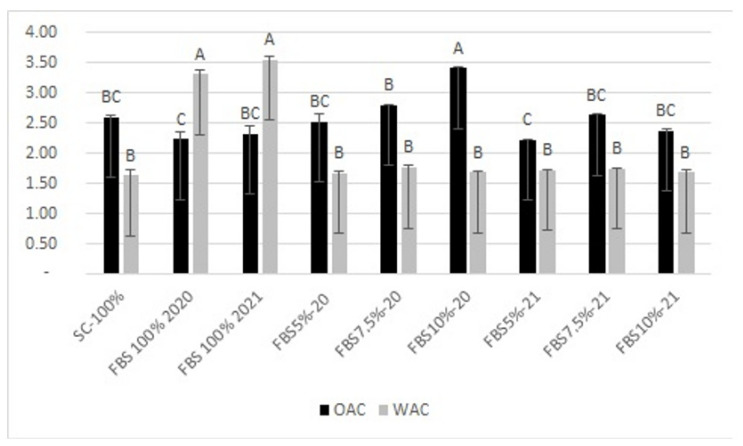
Water binding capacity (WBC; g water/g flour) and oil binding capacity (OBC; g oil/g flour) of re-milled semolina and of mixes prepared at increasing level of replacement (5, 7.5, 10%) with flour from ‘Signuredda’ bean. SC-100%: re-milled semolina 100%, i.e., control; FBS 2020: flour of bean ‘Signuredda’ 2020; FBS 2021: flour of bean ‘Signuredda’ 2021. Different letters indicate a significant difference (*p* ≤ 0.001).

**Figure 3 plants-12-01125-f003:**
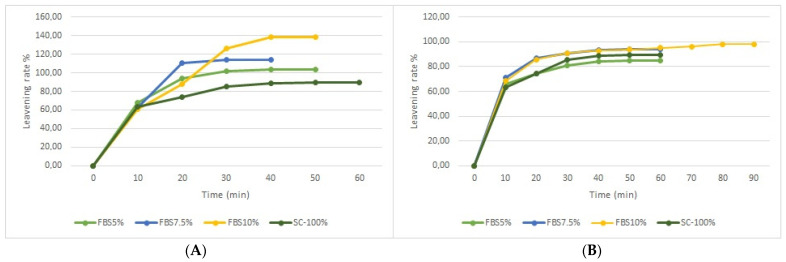
(**A**) Leavening rate (%) at increasing levels of replacement (5, 7.5, 10%) of re-milled semolina (SC-100%) with flour from ‘Signuredda’ bean 2020. (**B**) Leavening rate (%) at increasing levels of replacement (5, 7.5, 10%) of re-milled semolina (SC-100%) with flour from ‘Signuredda’ bean 2021.

**Figure 4 plants-12-01125-f004:**
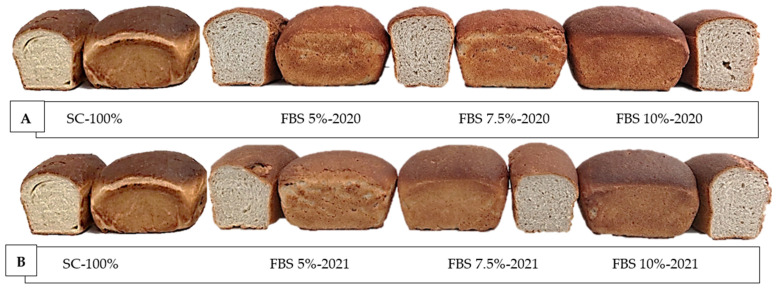
(**A**) Breads prepared, from left to right, with pure re-milled semolina and with flour mixes containing 5, 7.5, and 10% of flour from ‘Signuredda’ bean 2020. (**B**) Breads prepared, from left to right, with pure re-milled semolina and with flour mixes containing 5, 7.5, and 10% of flours from ‘Signuredda’ bean 2021.

**Table 1 plants-12-01125-t001:** Moisture and protein content of pure and mixes flour of ‘Signuredda’ bean in the two-year period 2020–2021.

Type of Flour	Moisture (g/100 g)	Protein (g/100 g)
*Pure flours*		
SC-100%	13.93 ± 0.13 a	11.90 ± 0.14 b
FBS 100%-2020	10.13 ± 0.13 e	18.81 ± 0.14 a
FBS 100%-2021	11.05 ± 0.05 de	19.20 ± 0.28 a
*Mixes*		
FBS 5%-2020	12.77 ± 0.16 bc	12.23 ± 0.04 b
FBS 7.5%-2020	12.26 ± 0.13 bc	12.40 ± 0.28 b
FBS 10%-2020	11.12 ± 0.14 de	12.57 ± 0.05 b
FBS 5%-2021	13.18 ± 0.05 ab	12.26 ± 0.06 b
FBS 7.5%-2021	11.91 ± 0.18 cd	12.43 ± 0.03 b
FBS 10%-2021	11.82 ± 0.24 cd	12.60 ± 0.28 b

FBS = flour of bean ‘Signuredda’; SC-100% = re-milled semolina 100%. Different letters in the columns indicate a significant difference (*p* ≤ 0.001).

**Table 2 plants-12-01125-t002:** Nutritional values of ‘Signuredda’ bean seeds in the two years 2020–2021 based on the INRAN (today CREA-AN, Rome, Italy) data for the raw bean reference.

Nutrient	‘Signuredda’ Bean 2020	‘Signuredda’ Bean 2021	Fresh Raw Bean Seed (INRAN Data)
Total carbohydrates (g/100 g)	43.1 ± 0.20 a	42.8 ± 0.14 a	47.5 ± 0.28 a
Total fats (g/100 g)	1.4 ± 0.11 d	1.3 ± 0.08 d	2.0 ± 0.42 d
of which saturated fatty acids (g/100 g)	0.3 ± 0.00	0.2 ± 0.01	-
Proteins (g/100 g)	18.8 ± 0.04 b	19.2 ± 0.02 b	23.6 ± 0.71 b
Dietary fiber (g/100 g)	17.1 ± 0.18 c	16.9 ± 0.10 c	17.5 ± 0.34 c
Salt (g/100 g)	0.006 ± 0.00 e	0.006 ± 0.00 e	0.006 ± 0.00 e
Energy value (Kcal)	294.4	293.5	291.0
Energy value (KJ)	1232	1228	1216

Different letters in the rows indicate a significant difference (*p* ≤ 0.001).

**Table 3 plants-12-01125-t003:** Mineral content of the different ‘Signuredda’ bean pure flours in the two years 2020–2021.

Mineral	Concentration (mg/100 g)	
	FBS 100%-2020	FBS 100%-2021
Calcium	478.3 ± 0.17 d	481.80 ± 0.30 d
Magnesium	698.74 ± 0.21 c	694.40 ± 0.88 c
Sodium	95.24 ± 0.88 e	97.84 ± 1.22 e
Potassium	6742.00 ± 7.07 a	6804.00 ± 7.07 a
Phosphorus	1839.00 ± 2.12 b	1826.00 ± 1.41 b
Copper	0.52 ± 0.02 g	0.65 ± 0.04 g
Manganese	2.39 ± 0.03 g	2.43 ± 0.20 g
Zinc	22.89 ± 0.08 f	23.37 ± 0.12 f
Iron	30.02 ± 0.03 f	28.76 ± 0.29 f

Different letters in the rows indicate a significant difference (*p* ≤ 0.001).

**Table 4 plants-12-01125-t004:** Color parameters of re-milled semolina (control), 100% bean flour, and mixes in the two years 2020–2021.

Type of Flour	Brown Index (100-L)	a*	b*
*Pure Flours*			
SC-100%	10.26 ± 0.01 f	−2.38 ± 0.00 e	17.17 ± 0.00 a
FBS 100%-2020	13.63 ± 0.07 b	1.08 ± 0.07 a	7.54 ± 0.08 f
FBS 100%-2021	14.49 ± 0.02 a	1.29 ± 0.02 a	8.38 ± 0.01 e
*Mixes*			
FBS 5%-2020	11.50 ± 0.01 e	−1.70 ± 0.02 d	16.72 ± 0.01 b
FBS 7.5%-2020	12.98 ± 0.04 c	−1.37 ± 0.03 bc	16.75 ± 0.01 b
FBS 10%-2020	12.20 ± 0.02 d	−1.23 ± 0.01 b	15.37 ± 0.01 d
FBS 5%-2021	11.49 ± 0.02 e	−1.77 ± 0.01 d	16.69 ± 0.02 b
FBS 7.5%-2021	12.74 ± 0.03 c	−1.57 ± 0.02 cd	16.05 ± 0.04 c
FBS 10%-2021	12.22 ± 0.03 d	−1.31 ± 0.05 b	15.33 ± 0.04 d

Different letters in the columns indicate a significant difference (*p* ≤ 0.001).

**Table 5 plants-12-01125-t005:** Color parameters: two-factor ANOVA (analysis of variance) of color parameters of 100% bean flour and mixes for the two years 2020–2021 (data are means ± standard deviations).

Year	Brown Index (100-L)	a*	b*
2020	12.58 ± 0.86 a	−0.80 ± 1.18	14.10 ± 4.09
2021	12.73 ± 1.18 a	−0.84 ± 1.32	14.11 ± 3.57

Different letters in the columns indicate a significant difference (*p* ≤ 0.001). If not indicated, it is ns (not significant).

**Table 6 plants-12-01125-t006:** Color parameters: two-factor ANOVA (analysis of variance) of color parameters of 100% bean flour and of mixes referring to the different integration percentages of flour (data are means ± standard deviations).

Integration Percentage	Brown Index (100-L)	a*	b*
*Pure Flours*			
FBS 100%	14.06 ± 0.50 a	1.18 ± 0.13 a	7.96 ± 0.49 c
*Mixes*			
FBS 5%	11.49 ± 0.02 d	−1.73 ± 0.05 d	16.70 ± 0.02 a
FBS 7.5%	12.85 ± 0.14 c	−1.47 ± 0.11 c	16.40 ± 0.41 b
FBS 10%	12.21 ± 0.02 b	−1.27 ± 0.05 b	15.35± 0.03 d

Different letters in the columns indicate a significant difference (*p* ≤ 0.001).

**Table 7 plants-12-01125-t007:** Main technological parameters of semolina and mixes in the two years 2020–2021: one-factor ANOVA (analysis of variance) (data are means ± standard deviations).

Type of Flour	Farinograph Data			Mixograph Data	
	Dough Development Time (min)	Stability (min)	Water Absorption at 500 B.U. * (%)	Mixing Time (min)	Peak Dough Height (M.U.) **
SC-100%	1.75 ± 0.07 a	3.15 ± 0.07 b	61.60 ± 0.14 b	2.65 ± 0.02 e	5.60 ± 0.14 bc
FBS 5%-2020	1.60 ± 0.00 ab	4.45 ± 0.07 a	66.55 ± 0.07 a	3.93 ± 0.02 bc	6.30 ± 0.14 b
FBS 7.5%-2020	1.45 ± 0.07 b	4.60 ± 0.14 a	67.30 ± 0.14 a	3.79 ± 0.01 c	5.05 ± 0.07 c
FBS 10%-2020	1.65 ± 0.07 ab	4.75 ± 0.07 a	68.30 ± 0.42 a	4.27 ± 0.02 a	4.90 ± 0.14 c
FBS 5%-2021	1.55 ± 0.07 ab	4.30 ± 0.14 a	66.50 ± 0.14 a	2.73 ± 0.01 e	7.65 ± 0.07 a
FBS 7.5%-2021	1.45 ± 0.07 b	4.60 ± 0.14 a	67.20 ± 0.28 a	3.40 ± 0.02 d	5.00 ± 0.14 c
FBS 10%-2021	1.65 ± 0.07 ab	4.75 ± 0.07 a	68.40 ± 0.71 a	4.10 ± 0.02 ab	5.05 ± 0.07 c

* Brabender units; ** Mixograph units. Different letters in the columns indicate a significant difference: dough development time *p* ≤ 0.05; stability and water absorption at 500 B.U. *p* ≤ 0.001; mixing time and peak dough height *p* ≤ 0.001.

**Table 8 plants-12-01125-t008:** Main technological parameters: two-factor ANOVA (analysis of variance) referring to the two years: 2020 and 2021 (data are means ± standard deviations).

Year	Farinograph			Mixograph	
	Dough Development Time (min)	Stability (min)	Water Absorption at 500 B.U. * (%)	Mixing Time (min)	Peak Dough Height (M.U.) **
2020	1.57 ± 0.10	4.60 ± 0.15	66.38 ± 0.81	3.99 ± 0.22 a	5.42 ± 0.69 b
2021	1.55 ± 0.10	4.55 ± 0.23	67.37 ± 1.00	3.41 ± 0.62 b	5.90 ± 1.35 a

* Brabender units; ** Mixograph units. Different letters in the columns indicate a significant difference (*p* ≤ 0.001). If not indicated, it is ns (not significant).

**Table 9 plants-12-01125-t009:** Main technological parameters: two-factor ANOVA (analysis of variance) referring to the different percentages of integration of bean flour (data are means ± standard deviations).

Integration Percentage	Farinograph			Mixograph	
	Dough Development Time (min)	Stability (min)	Water Absorption at 500 B.U. * (%)	Mixing Time (min)	Peak Dough Height (M.U.) **
FBS 5%	1.58 ± 0.05 ab	4.38 ± 0.13 ab	66.48 ± 0.05 a	3.33 ± 0.70 c	6.98 ± 0.78 a
FBS 7.5%	1.45 ± 0.06 b	4.60 ± 0.06 b	67.25 ± 0.06 a	3.59 ± 0.23 b	5.03 ± 0.10 b
FBS 10%	1.65 ± 0.06 a	4.75 ± 0.06 a	68.40 ± 0.06 a	4.18 ±0.10 a	4.98 ± 0.13 b

* Brabender units; ** Mixograph units. Different letters in the columns indicate a significant difference: Water absorption at 500 B.U. (*p* ≤ 0.001), Dough development time (*p* ≤ 0.05), Stability (*p* ≤ 0.01).

**Table 10 plants-12-01125-t010:** Evaluation of physical properties of durum wheat bread at increasing levels of replacement (5, 7.5, 10%) of re-milled semolina (control) prepared with flour of ‘Signuredda’ bean from different years: one-factor ANOVA (analysis of variance) (data are means ± standard deviations).

Type of Bread	Specific Volume (cm^3^/g)	Specific Weight (g/cm^3^)	Porosity *
SC-100%	3.05 ± 0.05 a	0.33 ± 0.01 ab	7.00 ± 0.00 a
FBS5%-2020	3.29 ± 0.00 a	0.30 ± 0.00 ab	5.00 ± 0.00 b
FBS7.5%-2020	3.25 ± 0.12 a	0.31 ± 0.01 ab	5.00 ± 0.00 b
FBS10%-2020	3.30 ± 0.18 a	0.30 ± 0.02 ab	5.50 ± 0.00 b
FBS5%-2021	2.97 ± 0.12 a	0.34 ± 0.01 a	5.50 ± 0.00 b
FBS7.5%-2021	3.40 ± 0.06 a	0.29 ± 0.00 b	6.00 ± 0.00 ab
FBS10%-2021	3.35 ± 0.13 a	0.30 ± 0.01 ab	5.25 ± 0.35 b

* Scale 1–8; 1 = non-uniform structure, large and irregular cells; 8 = uniform compact structure, small and regular cells. Different letters in the columns indicate a significant difference: specific volume and weight (*p* ≤ 0.05), and porosity (*p* ≤ 0.001).

**Table 11 plants-12-01125-t011:** Evaluation of physical properties of the bread samples produced using different levels of supplementation: two-factor ANOVA (analysis of variance) referring to the different percentages of integration of two years (data are means ± standard deviations).

Year	Specific Volume (cm^3^/g)	Specific Weight (g/cm^3^)	Porosity *
2020	3.28 ± 0.10	0.31 ± 0.01	5.17 ± 0.26 b
2021	3.24 ± 0.22	0.31 ± 0.02	5.58 ± 0.38 a

* Scale 1–8; 1 = non-uniform structure, large and irregular cells; 8 = uniform compact structure, small and regular cells. Different letters in the columns indicate a significant difference in porosity (*p* ≤ 0.01). If not indicated, it is not significant.

**Table 12 plants-12-01125-t012:** Evaluation of physical properties of the bread samples produced using different levels of supplementation: two-factor ANOVA (analysis of variance) referring to the different percentages of integration (data are means ± standard deviations).

Integration Percentage	Specific Volume (cm^3^/g)	Specific Weight (g/cm^3^)	Porosity *
FBS 5%	3.15 ± 0.17	0.32 ± 0.02	5.00 ± 0.00
FBS 7.5%	3.31 ± 0.10	0.30 ± 0.01	5.50 ± 0.58
FBS 10%	3.31 ± 0.15	0.30 ± 0.01	5.75 ± 0.29

* Scale 1–8; 1 = non-uniform structure, large and irregular cells; 8 = uniform compact structure, small and regular cells. If not indicated, it is not significant.

**Table 13 plants-12-01125-t013:** Colorimetric parameters of crust and crumb of durum wheat bread at increasing levels of replacement (5, 7.5, 10%) of re-milled semolina (control) prepared with ‘Signuredda’ bean flour from different years: one-factor ANOVA (analysis of variance) (data are means ± standard deviations).

	Crust			Crumb		
Sample	Brown Index (100-L)	a*	b*	Brown Index (100-L)	a*	b*
SC-100%	62.54 ± 1.10	12.85 ± 1.27	17.82 ± 4.11	23.78 ± 2.14 b	−2.92 ± 0.28 b	19.34 ± 2.04 a
FBS5%-2020	61.97 ± 1.51	14.95 ± 1.54	20.08 ± 1.73	28.79 ± 0.25 ab	−0.24 ± 0.17 a	10.83 ± 0.06 b
FBS7.5%-2020	61.93 ± 2.96	15.62 ± 0.35	20.88 ± 1.27	31.39 ± 0.40 a	0.09 ± 0.08 a	11.20 ± 0.28 b
FBS10%-2020	63.15 ± 3.18	14.80 ± 1.40	18.43 ± 3.38	31.06 ± 0.52 ab	0.40 ± 0.03 a	11.87 ± 0.02 b
FBS5%-2021	60.40 ± 0.02	15.00 ± 0.86	19.63 ± 0.54	26.59 ± 3.06 ab	−0.49 ± 0.01 a	12.30 ± 0.63 b
FBS7.5%-2021	59.93 ± 0.21	14.83 ± 1.39	20.37 ± 0.65	29.35 ± 2.59 ab	0.27 ± 0.49 a	11.06 ± 0.30 b
FBS10%-2021	66.05 ± 0.42	12.38 ± 0.18	14.03 ± 0.59	31.70 ± 1.74 a	0.75 ± 0.04 a	12.27 ± 0.12 b

Different letters in the columns indicate a significant difference, (*p* ≤ 0.05). If not indicated, it is not significant.

**Table 14 plants-12-01125-t014:** Evaluation of colorimetric parameters of the bread samples prepared with re-milled semolina and flour bean ‘Signuredda’ from different years, as determined by the two-factor ANOVA (analysis of variance) (data are means ± standard deviations).

	Crust			Crumb		
Year	Brown Index (100-L)	a*	b*	Brown Index (100-L)	a*	b*
2020	62.35 ± 2.15	15.12 ± 1.02	19.80 ± 2.11	30.41 ± 1.30	0.08 ± 0.30	11.30 ± 0.49 b
2021	62.13 ± 3.05	14.07 ± 1.51	18.01 ± 3.13	29.21 ± 3.01	0.18 ± 0.60	11.87 ± 0.70 a

Different letters in the columns indicate a significant difference (*p* ≤ 0.05). If not indicated, it is not significant.

**Table 15 plants-12-01125-t015:** Evaluation of colorimetric parameters of the bread samples prepared with re-milled semolina and flour bean ‘Signuredda’ at increasing levels of replacement (5, 7.5, 10%), as determined by the two-factor ANOVA (analysis of variance) (data are means ± standard deviations).

	Crust			Crumb		
Integration Percentage	Brown Index (100-L)	a*	b*	Brown Index (100-L)	a*	b*
FBS 5%	61.18 ± 1.26	14.98 ± 1.02	19.86 ± 1.08 ab	27.69 ± 2.18	−0.36 ± 0.17 b	11.56 ± 0.92 ab
FBS 7.5%	60.93 ± 2.06	15.22 ± 0.94	20.62 ± 0.87 a	30.37 ± 1.91	0.18 ± 0.31 ab	11.13 ± 0.25 b
FBS 10%	64.60 ± 2.50	13.59 ± 1.62	16.23 ± 3.22 b	31.38 ± 1.11	0.58 ± 0.20 a	12.07 ± 0.24 a

Different letters in the columns indicate a significant difference: b* crust and crumb (*p* ≤ 0.05), a* crumb (*p* ≤ 0.01). If not indicated, it is not significant.

**Table 16 plants-12-01125-t016:** Variation in physical characteristics of durum wheat bread at increasing levels of replacement (5, 7.5, 10%) of re-milled semolina (control) with flour from ‘Signuredda’ bean from different years, during 7 days of storage.

Days After Baking	Type of Bread	Moisture (g/100 g)	Volume (cm^3^)	Weight (g)	Height (mm)	Hardness (N)
T0	SC-100%	31.5 ± 2.12 b(a)	440.0 ± 3.54 b(a)	144.5 ± 1.13 b(a)	82.0 ± 1.20	12.1 ± 0.64 a(c)
	FBS5%-2020	37.5 ± 0.71 a(a)	480.0 ± 0.00 a(a)	146.0 ± 0.18 ab(a)	78.7 ± 0.21	9.0 ± 1.33 ab(b)
	FBS7.5%-2020	36.0 ± 1.41 a(a)	485.0 ± 17.68 a(a)	149.4± 0.21 a(a)	78.0 ± 1.91	6.8 ± 1.03 b(b)
	FBS10%-2020	35.5 ± 0.01 ab(a)	480.0 ± 28.28 a	145.5 ± 0.67 ab(a)	79.9 ± 4.24	7.5 ± 0.23 b(b)
	FBS5%-2021	34.5 ± 0.71 ab(a)	437.5 ± 10.61 b(a)	147.1 ± 2.37 ab(a)	74.1 ± 4.67	9.2 ± 1.64 ab(b)
	FBS7.5%-2021	35.0 ± 0.00 ab(a)	497.5 ± 16.61 a(a)	146.4 ± 0.71 ab(a)	80.0 ± 3.04	7.0 ± 0.97 b(c)
	FBS10%-2021	35.5 ± 0.71 ab(a)	487.5 ± 17.68 a(a)	145.5 ± 0.25 ab(a)	82.9 ± 0.49 (a)	8.3 ± 0.85 ab(b)
T1	SC-100%	27.5 ± 0.71 (ab)	408.2 ± 3.28 b(ab)	134.4 ± 1.05 b(ab)	79.9 ± 1.17	33.7 ± 0.03 a(ab)
	FBS5%-2020	27.5 ± 2.12 (ab)	445.3 ± 0.35 ab(b)	141.0 ± 0.74 ab(ab)	76.5 ± 3.32	12.5 ± 2.35 b(ab)
	FBS7.5%-2020	30.0 ± 0.00 (ab)	462.5 ± 21.21 a(a)	144.6 ± 0.00 a(ab)	75.9 ± 1.56	9.7 ± 2.13 b(ab)
	FBS10%-20	31.0 ± 0.00 (ab)	462.3 ± 28.64 a	141.8 ± 0.14 a(a)	78.1 ± 3.46	11.3 ± 1.76 b(b)
	FBS5%-2021	28.0 ± 1.41 (b)	412.5 ± 7.07 b(ab)	140.4 ± 1.22 ab(ab)	71.4 ± 3.39	14.3 ± 0.94 b(b)
	FBS7.5%-2021	30.0 ± 1.41 (ab)	467.3 ± 6.72 a(a)	138.5 ± 1.28 ab(ab)	78.9 ± 1.56	11.1 ± 1.87 b(bc)
	FBS10%-2021	29.5 ± 0.71 (a)	462.8 ± 24.40 a(ab)	138.6 ± 0.22 ab(ab)	78.9 ± 0.35 (ab)	14.7 ± 3.29 b(b)
T4	SC-100%	26.0 ± 0.00 (b)	400.2 ± 3.22 abc(ab)	124.4 ± 0.97 ab(bc)	78.7 ± 1.15	28.2 ± 0.24 (b)
	FBS5%-2020	28.5 ± 3.54 (ab)	380.8 ± 1.06 bc(c)	127.6 ± 3.25 ab(bc)	75.5 ± 4.67	23.4 ± 1.27 (ab)
	FBS7.5%-2020	30.0 ± 4.24 (ab)	390.0 ± 14.14 bc (b)	129.7 ± 2.55 a(bc)	75.1 ± 2.12	24.7 ± 0.13 (a)
	FBS10%-2020	26.0 ± 4.24 (b)	440.0 ± 21.21 ab	129.5 ± 0.09 a(b)	77.0 ± 3.89	22.0 ± 2.67 (ab)
	FBS5%-2021	25.5 ± 0.71 (b)	360.0 ± 7.07 c(bc)	129.2 ± 0.07 a(ab)	70.7 ± 2.40	28.3 ± 0.76 (ab)
	FBS7.5%-2021	31.0 ± 0.00 (a)	460.0 ± 7.07 a(a)	122.3 ± 0.30 b(bc)	77.9 ± 0.92	25.3 ± 3.35 (ab)
	FBS10%-2021	31.0 ± 0.00 (a)	412.5 ± 17.68 abc(b)	122.4 ± 0.05 b(bc)	77.9 ± 0.28 (b)	23.7 ± 1.84 (ab)
T6	SC-100%	25.0 ± 0.00 a(b)	386.3 ± 3.10 ab(b)	119.4 ± 0.93 (c)	77.5 ± 1.14	39.0 ± 1.00 (a)
	FBS5%-2020	22.5 ± 0.71 ab(b)	351.0 ± 1.41 c(d)	120.3 ± 3.08 (c)	74.3 ± 3.89	37.2 ± 3.26 (a)
	FBS7.5%-2020	22.5 ± 3.54 ab(b)	383.8 ± 8.84 abc(b)	124.0 ± 5.42 (c)	73.4 ± 1.34	26.9 ± 1.44 (a)
	FBS10%-2020	24.0 ± 0.00 ab(b)	392.5 ± 3.54 a	122.0 ± 0.47 (c)	76.2 ± 3.68	30.6 ± 4.85 (a)
	FBS5%-2021	24.5 ± 0.71 a(b)	355.0 ± 7.07 bc(c)	122.9 ± 2.32 (b)	69.9 ± 1.84	35.3 ± 2.57 (a)
	FBS7.5%-2021	22.0 ± 0.00 ab(b)	403.8 ± 8.84 a(b)	119.8 ± 2.42 (c)	76.7 ± 0.42	32.0 ± 3.75 (a)
	FBS10%-2021	18.5 ± 0.71 b(b)	394.5 ± 7.78 a(b)	119.1 ± 5.15 (c)	77.4 ± 0.14 (b)	32.6 ± 3.42 (a)

SC-100% = re-milled semolina 100%, i.e., control; FBS 2020: flour of bean ‘Signuredda’ 2020; FBS 2021: flour of bean ‘Signuredda’ 2021. Different lower-case letters in a column indicate a significant difference (*p* ≤ 0.05) among different types of bread within the same day. Different lower-case letters in brackets in a column indicate a significant difference (*p* ≤ 0.01) for the same type of bread at different storage times. If not indicated, it is ns (not significant).

**Table 17 plants-12-01125-t017:** Formulation of the experimental breads (g for 100 g of re-milled semolina).

Bread Type	Re-Milled Semolina	FBS 2020	FBS 2021	Yeast	NaCl	Ascorbic Acid	Sugar	Shortening	Water *
SC-100%	100	-	-	0.6	0.4	8 × 10^−4^	1.2	3.5	61.6
FBS 5%-2020	95	5	-	0.6	0.4	8 × 10^−4^	1.2	3.5	66.6
FBS 7.5%-2020	92.5	7.5	-	0.6	0.4	8 × 10^−4^	1.2	3.5	67.3
FBS 10%-2020	90	10	-	0.6	0.4	8 × 10^−4^	1.2	3.5	68.3
FBS 5%-2021	95	-	5	0.6	0.4	8 × 10^−4^	1.2	3.5	66.5
FBS 7.5%-2021	92.5	-	7.5	0.6	0.4	8 × 10^−4^	1.2	3.5	67.2
FBS 10%-2021	90	-	10	0.6	0.4	8 × 10^−4^	1.2	3.5	68.4

SC-100% = re-milled semolina 100%, e.g., control; FBS = flour of ‘Signuredda’ bean. * Amount corresponding to the farinograph water absorption at 500 B.U.

## Data Availability

All available data are reported in the paper.

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
