# Peer review of "Effects of Partial Replacement of Durum Wheat Re-Milled Semolina with Bean Flour on Physico-Chemical and Technological Features of Doughs and Breads during Storage"

_plants, 2023, doi:10.3390/plants12051125_

Round 1

Reviewer 1 Report

The manuscript presents the physico-chemical and technological properties of doughs and durum wheat re-milled breads supplemented with bean flour at 0%, 5%, 7.5% and 10%. The manuscript is presented in a well-structured manner. The manuscript is very concrete and with very complete information. Such study is important in the field of food science and shows that there is a potential for producing functional bread with superior nutritional quality. The following points commented on in the attached manuscript should be considered.

Author Response

Reviewer 1

  1. I would recommend You specify values here, e.g. an increase in water absorption from..... to the level .....

Answer: We thank you for your review. According to your suggestion we modify the sentence in lines 26-28 as follows:

The water absorption and dough stability on the farinograph increased from 1.45, in FBS 7.5%, to 1.65 in FBS 10%, for both 2020 and 2021, for water absorption. Dough stability increased from 4.30 in FBS 5%-2021 to 4.75 in FBS 10%-2021.

  1. Please check the grammatical correctness of this verb

Answer: Thanking you for your review, we modified the sentence in line 33:

The fermentation test with the addition of 10% bean flour significantly increased the fermentative capacity of the dough.

  1. Please pay attention to whether the given significance of differences in a given feature is correct. In my opinion, the letters a, b, c ... should be given in relation to e.g. total carbohydrates, total fats et.

Answer: We have accepted your suggestion. We corrected the data in the TABLE 2 (line 118) by performing a new statistical analysis.

  1. The letters a,b,c... are given after a space, while in table 1 without space. Please unify so that all tables are the same.

Answer: Following your suggestion, we have arranged the letters of table 1 (line 100) and table 3 (line 126), inserting a space between the standard deviation and the letters.

  1. As in table number 2.As You analyze the significance of differences regarding the content of calcium

Answer: We have arranged table 3 (line 126) by inserting a space between the standard deviation and the letters concerning the statistical analysis.

  1. As in abstract I would recommend You specify values here, without this sentence.

Answer: We thank you for your review. As you suggested, we corrected the sentence in lines 191-192 by inserting the values.

Water absorption increased from 1.45, in FBS 7.5%, to 1.65 in FBS 10%.

  1. Please insert space after the percentage

Answer: Thank you for pointing out this error. We proceeded to insert a space after the percentage in line 191.

SC-100% sample.

  1. Please consider to provide specific values here, to what level of the water absorption has increased

Answer: Following your report, we have improved the sentence referring to the Hoxha et al. (lines 198-201)

This positive trend of increased water absorption following an increase in the percentage of bean flour supplementation has been found by other authors. Hoxha et al. [30] recorded a similar trend, following the integration of bean flour from 10% to 15%, with values ranging respectively from 56.1% to 57.2%.

  1. Table 10 concerns the physical properties of bread, so I suggest changing the description to be bread not flour

Answer: In agreement with your request, we have changed the description from “Type of flour” to “Type of bread”, in the table 10 (line 354).

  1. Please insert the space and capital letter (Table 13)

Answer: Thanks for your suggestion, we have corrected the sentence (Table 13) in the line 420.

  1. The colorimetric parameters of crust are missing letters denoting significant differences. Please complete it.

Answer: The absence of letters in the colorimetric parameters of the bread crust in table 13 (line 443) is due to the absence of statistically significant differences. A specific note has been inserted below the table (lines 447-448).

  1. Please round the values of the tested parameters to the decimal place.

Answer: We thank you for your suggestion. Following your indication, we have rounded the values of the tested parameters in the table 16 (line 536) to the decimal place.

  1. Closing parenthesis is missing

Answer: Thanks for your suggestion, we've added the missing parenthesis (line 554).

  1. Please insert the correct degree sign as in line 622

Answer: Thanks for your suggestion, we've added the correct degree sign in line 582

  1. Please insert the correct degree sign as in line 622

Answer: Thanks for your suggestion, we've added the correct degree sign in line 589.

  1. If the word bread is given in the title, I believe that the ingredients of the bread recipe and partly a description of the baking process should be given.

Answer: Thanks for your suggestion. As you requested, we have added the ingredients of the recipe (table 16, line 639) and the description of the baking process (lines 627-637).

Reviewer 2 Report

The manuscript has investigated the physico-chemical and technological features of doughs and breads during storage, which are prepared by partial replacement of durum wheat re-milled semolina with bean flour. The topic is interesting and the results are well discussed. However, it has several problems:

1. Language should be edited by a professional English editor.

2. L 34-36; This sentence is a possible conclusion and was not evaluate in this work. So, I suggest the authors to change the sentence. 

3. Please give more details about the genotype "Signuredda" and why this type was chosen.

4. L 89; Please write "Table 1" instead of "Tab. 1".

5. Figure 2; Please check the standard deviations and the significant letters.

6. Table 13; Please add the significant letters.

Author Response

Reviewer 2

  1. The manuscript has investigated the physico-chemical and technological features of doughs and breads during storage, which are prepared by partial replacement of durum wheat re-milled semolina with bean flour. The topic is interesting and the results are well discussed.

Answer: We thank the Reviewer for appreciating our work.

  1. Language should be ed-ited by a professional English editor.

Answer: Thanks for the review. We have provided a new language edit, by a professional English editor.

  1. L 34-36; This sentence is a possible conclusion and was not evaluate in this work. So, I suggest the authors to change the sentence.

Answer: Thanks for your review. Following your suggestion, we changed the sentence, in lines 36-38, as follows:

In conclusion, the results showed an interesting potential of 'Signuredda' bean flour as a bread-making ingredient to obtain softer breads, which better resist stale.

  1. Please give more details about the genotype "Signuredda" and why this type was chosen.

Answer: In response to your request, we included a description of the bean genotype used (lines 563-576).

The 'Signuredda' bean is a local variety of Phaseolus vulgaris L. with a determined growth and a creamy-white pod. The seeds are reniform with a firm, nut-coloured, white ileum around which there is a reddish ring. The cotyledons are light yellow. It is a bean that cooks easily, even if it doesn’t need to be soaked. It is very mellow and tasty and has various uses: soups, in salads after boiling, ground to obtain flour, etc. Although it does not have a brown integument, it leaves the cooking water dark in colour. It is a genotype with a strong bond with the territory of origin (north-eastern Sicily, in the province of Messina), where it is grown in a still restricted area.

The authors decided to use this local variety of bean because it is easily grindable, thanks to its slightly waxy, light-yellow cotyledons which give rise to a flour very similar in color to that of semolina, and its firm tegument which shatters easily. Furthermore, based on the results of preliminary rheological and technological tests on the doughs of the mixes with durum wheat semolina, it was seen that it had good potential for making bread.

  1. L 89; Please write "Table 1" instead of "Tab. 1".

Answer: Thanks for your suggestion. We have changed Tab.1 with Table 1, in line 91.

  1. Figure 2; Please check the standard deviations and the significant letters.

Answer: Thanks for your review. We checked Figure 2 (line 290) and performed a new statistical analysis. Errors have been corrected.

  1. Table 13; Please add the significant letters.

Answer: The absence of letters in the colorimetric parameters of the bread crust in table 13 (line 443) is due to the absence of statistically significant differences. A specific note has been inserted below the table (lines 447-448).

Reviewer 3 Report

The introduction section is very well-detailed, and the aim of the work is well-defined. I appreciate that the authors have carried out research on the effect of other types of flour on bakery products, from the specialized literature, also using new references.

line 55 - The authors wrote that many publications emphasize the importance of adding different types of flour to enhance the nutritional value of wheat flour. The authors should cite here more than one publication [8]. I suggest the following up-to-date literature: https://doi.org/10.3390/agronomy12010137; https://doi.org/10.1038/s41598-022-12017-7

Results and discussion sections are well related and important scientific information is presented.

Table 3: Insert „.” At Calcium St. Dev. value for FBS 100% - 2021 and insert spaces between numbers and letters for the first column.

Table 5 is missing the letter for the significant differences in the a* and b* values. Same remark for table 8, for DDT, Stability, and WA; for tables 11 and 12, I did not identify the differences between parameters.  For porosity has to be at uppercase *, not a?

Line 439,462: Tables 13 and 14: There are not inserted different letters in columns, which have to indicate the significant difference, and table 15 there are presented only partial differences. Please revise.

In Figure 3, why the authors decide to use coma in the B figure, on the OX axis?

The methods are well-described and can be easily reproduced. The authors strictly followed all analysis methods.

Author Response

Reviewer 3

  1. The introduction section is very well-detailed, and the aim of the work is well-defined. I appreciate that the authors have carried out research on the effect of other types of flour on bakery products, from the specialized literature, also using new references.

Answer: We thank the reviewer for appreciating our work.

  1. Line 55 - The authors wrote that many publications emphasize the importance of adding different types of flour to enhance the nutritional value of wheat flour. The authors should cite here more than one publication [8]. I suggest the following up-to-date literature: https://doi.org/10.3390/agronomy12010137; https://doi.org/10.1038/s41598-022-12017-7

Answer: We thank the Reviewer. As suggested, we have added the proposed references (line 58)

  1. Results and discussion sections are well related and important scientific information is presented.

Answer: We thank the reviewer for appreciating our work.

  1. Table 3: Insert „.” At Calcium St. Dev. value for FBS 100% - 2021 and insert spaces between numbers and letters for the first column.

Answer: We thank the reviewer for the reports. We have corrected all the requests regarding table 3.

  1. Table 5 is missing the letter for the significant differences in the a* and b* values. Same remark for table 8, for DDT, Stability, and WA; for tables 11 and 12, I did not identify the differences between parameters. For porosity has to be at uppercase *, not a?

Answer: Thank you for the reports. In tables 11 and 12 we have corrected the letter a with * in the porosity column. The absence of letters in tables 5, 8, 11, 12, is due to the absence of statistically significant differences. For greater clarity we have added a specific note below each of the affected tables.

If not indicated, it is ns (not significant).

  1. Line 439,462: Tables 13 and 14: There are not inserted different letters in columns, which have to indicate the significant difference, and table 15 there are presented only partial differences. Please revise.

Answer: Thank you for the reports. The absence of letters in tables 13, 14, 15 is due to the absence of statistically significant differences. For greater clarity we have added a specific note below each of the affected tables.

If not indicated, it is ns (not significant).

  1. In Figure 3, why the authors decide to use coma in the B figure, on the OX axis?

Answer: We thank the reviewer for the report. We have corrected the errors in the figure 3B.

  1. The methods are well-described and can be easily reproduced. The authors strictly followed all analysis methods.

Answer: We thank the reviewer and are pleased that he appreciated our work.
